# Valorization of Lignin as an Immobilizing Agent for Bioinoculant Production using *Azospirillum brasilense* as a Model Bacteria

**DOI:** 10.3390/molecules24244613

**Published:** 2019-12-17

**Authors:** Victor Rogelio Tapia-Olivares, Eimy Alejandra Vazquez-Bello, Efrén Aguilar-Garnica, Froylán M.E. Escalante

**Affiliations:** Department of Biotechnology and Environmental Sciences, Universidad Autónoma de Guadalajara, AC. Av. Patria 1201, Lomas del Valle, 45129 Zapopan, Mexico; vtaov4@gmail.com (V.R.T.-O.); eimy98@live.com.mx (E.A.V.-B.); efren.aguilar@edu.uag.mx (E.A.-G.)

**Keywords:** organosolv, biofertilizer, nitrogen-fixing bacteria, *Azospirillum*

## Abstract

Plant growth-promoting bacteria (PGPB) have been largely considered as beneficial in harsh and limiting environments given their effects on alleviating plant stress. For practical applications, most of the PGPB are prepared in immobilization matrices to improve the stability and benefits of bacteria. Despite the long list of immobilizing agents/carriers tested to date, a long list of desired requirements is yet to be achieved. Here, lignin stands as a scarcely tested immobilizer for bioinoculants with great potential for this purpose. The aim of this work was to demonstrate the feasibility of lignin as a carrier of the nitrogen-fixing *Azospirillum brasilense*. These bacteria were cultured in liquid media with recovered organosolv lignin added for bacterial immobilization. Then, lignin was recovered and the immobilized biomass was quantified gravimetrically by DNA extraction and serial dilution plating. Fluorescent microscopy as well as Congo red agar plating showed the immobilization of the bacterial cells in the lignin matrix and crystal violet dyeing showed the biofilms formation in lignin particles. A high number of cells were counted per gram of dried lignin. Lignin can be readily used as low-cost, health-safe bioinoculant carrier to be used in soil and agricultural applications.

## 1. Introduction

The use of plant growth-promoting bacteria (PGPB) as agriculture inoculants is extensive, given their multiple benefits reported as aids for revegetation, reforestation, increase of crop production, pathogen control, and soil amendment [1,2]. PGPB include a wide range of soil microorganisms, with some of them in symbiotic association with their host plant, like *Rhizobium* spp., fixing atmospheric nitrogen, while some non-symbiotic PGPB are associated with their host plant, such as *Pseudomonas* spp., *Bacillus* spp., *Azospirillum* spp., and *Burkholderia* spp. [2]. In some cases, these PGPB are directly inoculated in crop seeds; however, these have a short viability, which make it necessary to develop formulations able to sustain microbial growth, to resist drying and rehydration, and, of course, to deliver living microorganisms to the soil [1]. Therefore, such formulations, also called bioinoculants, are prepared in immobilizing agents or carriers. Cell immobilization provides several productivity advantages. In the specific production of bioinoculants, it enables the possibility of simplifying the biomass recovery from culture media and to preserve viable cells.

The main requisites for bioinoculants are: (a) to promote the microbial growth in soil, (b) to preserve cell viability, and (c) to provide sufficient microorganisms to generate a physiological plant response [3]. There are several materials used to prepare bioinoculants, namely abiotic substrates, which can be in solid, liquid, or gel forms [4], which can be categorized as: (a) residues from plants and soil, such as turf, clay, lignin, bagasse, sawdust, charcoal, and compost; (b) inert materials, including polymers, vermiculite, and perlite; and (c) liquids, which are added to the culture media to improve adherence, stability, and microbial dispersion [1,5]. An extended list of requirements and particularities for the carriers have been previously reported [1,6]. In general, the selection of the immobilizing agent depends upon the carrier’s material stability, mechanical strength, lack of toxicity, and the cost of the material [7,8].

In the search for more accessible materials for bioinoculants, lignin appears to be a feasible option. Lignin is an abundant and intrinsic component of the lignocellulosic biomass, it is innocuous, economic, and, most of all, a large renewable organic resource present in plants [9,10,11]. For many years, lignin has been recovered as a pulp and paper industry byproduct (50 million tons annually) and, more recently, with the continued growth in global cellulosic ethanol production, larger quantities of residual lignin are being generated (1.0 kg/L of cellulosic ethanol). In 2017, an estimated 70 million tons of lignin were produced, given its low value, most of it was burnt for energy production [12]. Considering the current increasing generation of lignin, it is an ideal and suitable carrier for bioinoculant production due to its organic nature and its inert matrix capabilities.

From the above mentioned PGPB, the genus *Azospirillum* seems to be one of the most studied, with hundreds of studies related to its use. Apart from its stimulatory action in plant nutrient absorption, *Azospirillum* spp. enhance the root mass formation, providing physical stability to the plant against wind and water erosion and resistance to abiotic stress. Moreover, *Azospirillum* spp. have the ability to grow under extreme pH and heavy metal conditions [13]. Given its importance in agriculture, this genus has been used in different formulations for producing inoculants, from free cells to immobilized ones, and the main carrier materials used have been peat, poultry manure, soil, turf, clay, and alginate [1]. The aim of this work was to demonstrate the feasibility of lignin as a carrier of nitrogen-fixing *Azospirillum brasilense*.

## 2. Results

### 2.1. Characteristics of Lignin

FTIR spectra for lignin used in this study are shown in Figure 1. The spectra have been originally recorded in the range between 4000 and 400 cm^−1^, but are presented in the range between 1800 and 700 cm^−1^ as this range contains the most remarkable peaks for lignin. Particularly, the peak at 1030–1035 is an indicator of aromatic C–H in-plane deformation plus C–O deformation; the peak at 1120 cm^−1^ is attributed to a secondary alcohol and C=O stretching, which is typical of a syringyl unit [14]; while the peak at 1210 cm^−1^ is related to the C–O vibrations of primary alcohols [15]. Finally, the peak at 1422–1430 cm^−1^ is an indicator of aromatic skeleton vibrations combined with C–H in-plane deformations [14]. These characteristics are reported since lignins, which were extracted with other solvents or technologies, may present different results given their particular structures [16]. As previously reported, the bands at 1225 and 1034 cm^−1^ indicated the presence of both guaiacyl and syringyl unit in the lignin molecule [14].

### 2.2. Cell Immobilization in Lignin Particles

Lignin particles recovered from the bacterial culture media were plated in Congo red agar after washing and drying them to verify the presence of *Azospirillum brasilense*. In general, the particles generated red colonies indicating the presence of the bacteria. Then, other lignin particles were collected and dyed using Acridine orange colorant for observation under the fluorescent microscope. There, it was possible to identify live adhered cells in the lignin surface, as shown in Figure 2. Even when some cell bacteria were moving in the background, those cells in the lignin particle remained static, which is indicative of effective immobilization (Figure 2a). Also, in another lignin particle it was observed that those cells in the surface of the lignin were alive (green cells) and the rest were unviable cells (orange cells, Figure 2b).

### 2.3. Cell Enumeration in Lignin Particles

In order to quantify the cell density per gram of lignin particles, three bacterial enumeration techniques were assayed and correlated. Given the difficulties in cell enumeration on adhered living cells in the lignin surface, free bacterial cells were dry-weighed and correlated with extracted DNA. Since this method does not differentiate among living and dead cells, the serial dilution plating technique was applied and the viable cell number was also correlated with the extracted DNA to estimate the number of living cells in the lignin particles. A final concentration of 2.85 × 10^9^ cells/g of lignin was estimated, which represents a bacterial cell recovery from the liquid medium of 1.25 ± 0.14%. This led us to consider that each liter of medium can have 80 g of lignin added in order to recover the maximum number of viable bacteria.

### 2.4. Biofilm Formation in Lignin Particles

Biofilm formation was quantitated by cystal violet (CV) dyeing assay using three different culture media, including nitrogen-rich LB (Luria–Bertani) medium, nitrogen-source minimal medium OAB [1], as well as minimal medium without nitrogen source. *A. brasilense* could form biofilms in all tested media, mostly in OAB+ (with ammonium chloride) where biofilm biomass was kept constant. When growing in OAB− (without nitrogen source), the biofilm production increased up to day 3 and then decreased. When growing LB, biofilm accumulation grew exponentially up to day 4 and then remained constant. When comparing the maximum biofilm formation between media, OAB+ was 1.35-times higher than OAB− (at day 3) and 1.1-times higher than LB (at day 5, Figure 3).

## 3. Discussion

The intention of this work was to produce a bioinoculant to be applied in the agricultural field using a novel carrier for the nitrogen-fixing *Azospirillum brasilense*. This rhizobacteria is considered to be one of the most studied and, therefore, is highly used for plant-growth promotion applications. It was decided to explore the feasibility of lignin as the inert support for this bacterium, taking advantage of recent discussions about the need for lignin valuation. Lignin is already generated in high amounts in the paper and pulp industry and, more recently, lignin production has increased as a residue from second-generation biofuels, mainly cellulosic ethanol. Due to the multiple properties of lignin, such as availability, porosity, low cost, thermal stability, nontoxicity, and biocompatibility, it is a good source for innovative “green” applications [17]. It was hypothesized that lignin could be used as efficient immobilizer of plant-growth promoting bacteria *A. brasilense*.

Immobilization presents many advantages over free-cell systems; it makes cells more tolerant to changing environmental conditions and less vulnerable to toxic substances, and adsorption is one of the most used techniques for cell immobilization. Furthermore, it has been reported that adhesion to surfaces makes bacterial communities more active than their free-living counterparts; adhesion frequently increases in the exponential growth phase given the increment in the cell wall hydrophobicity [18]. In this study, we took advantage of the ability of *A. brasilense* to form biofilms and its consequent adherence to prepare a bioinoculant with organosolv-recovered lignin. We tracked the biofilm formation on lignin particles in different culture media [19,20]. We observed a good adhesion of bacterial particles to lignin as shown before; this could be explained in many ways, but we can attribute the facility of adherence to the fact that *A. brasilense* forms more stable biofilms in hydrophobic surfaces [20], which is the case of lignin. Also, it seems that lack of nitrogen negatively affected the biofilm formation after 3 days, after which they started to diminish in relative biomass; different from *Pseudomonas stutzeri* A1501, which forms more biofilms under nitrogen-depleted conditions [19].

Lignin has been previously reported as a carrier of microorganisms to be used in biofertilizer production with *Azotobacter vinelandii* [21] as a feasible carrier for this nitrogen-fixing bacteria. Also, lignin has been used as carrier of the microbiome for anaerobic digestion [17]. We proposed here the use of *A. brasilense* immobilized in lignin, taking advantage of the ability of this bacteria to form biofilms and remediate degraded soils as well as the possibility to reincorporate a plant residue to biogeochemical cycles. This could help to value lignin and increase organic matter and microbial content of eroded soils.

A few decades ago, Bashan [22] suggested the necessity for high cellular density in bioinoculants based on *Azospirillum* to assure its soil activity, i.e., a minimal concentration of 1 × 10^6^–1 × 10^7^ cells per plant must be inoculated. In 1996, Fallik and Okon compared bentonite, talc powder, and basalt to immobilize *A. brasilense*, achieving cell densities ranging from 3 × 10^7^ to 1.8 × 10^9^ cells/g of carrier, respectively. In more recent reports, Piccini et al. [23] treated wheat seeds with peat inoculant containing 1 × 10^8^ cells/g, Bashan et al. [24] reported cell densities higher than 1 × 10^7^ cells/alginate bead, and Schoebitz et al. [25] tested several combinations of alginate to prepare bioinoculant beads with *A. brasilense*, achieving cell densities around 1 × 10^9^ cells/g of bead. Also, when turf was used as carrier of *A. brasilense*, the cellular density ranged from 5 × 10^7^ to 5 × 10^8^ cells/g [20]. The results presented in this paper (2.85 × 10^9^ cells/g of lignin) agrees with that reported by other authors, indicating that lignin can be an appropriate carrier of bacterial cells for bioinoculants production. Even when we have not tested the efficiency of our bioinoculant in the soil or with plants, we considered that using lignin for this purpose would have similar results to those previously observed with the advantage that we are returning that lignocellulosic material to the agricultural land. If it is considered that the lignin used in this study came from the residues generated during the production of bioethanol using wheat straw as the raw material, the use of this lignin in agricultural or eroded soils would close, at least in part, a continuous productive cycle.

## 4. Materials and Methods

### 4.1. Lignin Extraction and Characterization

Lignin was recovered from residual wheat straw coming from the fermentation stage of a pilot-scale biochemical platform biorefinery by the organosolv method. The residual wheat straw was washed with water, as previously described [14], and then with ethanol-water 50% *w*/*w*. The organosolv process for lignin extraction was carried out in a pressurized reactor (Labscale tailor-made) using a mass-solvent ratio 1:15 (*w*/*v*) under agitation at 250 rpm with 30 mM sulfuric acid as a catalyst at 160 °C for 30 min; the solvent was ethanol-water 50% *w*/*w*. The solid product was washed again with a solvent solution and then dried. Finally, the recovered lignin was characterized via FTIR spectroscopy (Thermo Scientific, Waltham, MA, USA). FTIR spectra were obtained with 16 scans at a resolution of 4 using a Thermo Scientific Nicolet is5 FTIR spectrometer.

### 4.2. Growth and Immobilization of Azospirillum Brasilense

*A. brasilense* was conserved in agar plates with Luria–Bertani (LB) medium added with ampicillin (100 µg/L). Bacterial colonies collected from these agar plates were pre-inoculated in liquid LB up to OD_540_ = 0.05. Then, 5 mL of the culture were transferred to 500 mL Erlenmeyer flasks with 200 mL of LB added with 100 mg of organosolv extracted lignin. All media, including those containing lignin, were wet heat sterilized under standard conditions. The lignin-added LB was incubated at 32 ± 2 °C for 24 h with continuous agitation at 120 rpm. After the first 24 h, the culture was left static for an additional 72 h to promote the adherence of *A. brasilense* to the lignin particles, (See Appendix A). Finally, the liquid medium was decanted, and the lignin particles were recovered by centrifugation at 4000 *g* for 10 min. The remaining solid was washed thrice with physiological saline solution (0.85% *w*/*v*) and centrifuged at the same conditions prior to being oven dried at 60 ± 1 °C for 4 h. Samples of the lignin particles were spread in Congo red agar plates to verify bacterial immobilization. Lignin particle samples were dyed using orange Acridine solution and observed in an epifluorescence microscope (Zeizz, Germany) for bacterial cell adherence confirmation.

### 4.3. Cell Quantification

Quantification of *A. brasilense* was carried out in three ways: gravimetrically, by serial dilution plate, and DNA extraction techniques. DNA extraction was carried out as described by Andreou [26]. Briefly, a gravimetric standard curve and serial dilution curves were prepared with free cells to correlate the biomass (mg) and cell density (no. of cells/mL) with extracted DNA (µg/mL). To compute the cell density in lignin particles (DW biomass in mg/g of lignin or no. cells/g lignin) the amount of extracted DNA was extrapolated to corresponding cell density.

### 4.4. Biofilm Assay

Biofilm biomass quantification was carried out, as previously described, with slight modifications [19,27]. Following the specifications mentioned above, *A. brasilense* was grown in 125 mL Erlenmeyer flasks containing 50 mL of one of three different media: LB, OAB with nitrogen source (OAB+), or OAB without nitrogen (OAB−) [28] and 50 mg of lignin. All cultures were set at 30 ± 1 °C and 100 rpm for 24 h, then left at rest for the rest of the experiment. At days 2, 3, 4, and 5, lignin was recovered from each flask by filtration in Whatman paper and then washed thrice with saline solution (NaCl, 0.85 % *w*/*v*). All samples were dried at 60 °C for 2 to 3 min. Then, two aliquots of 10 mg of dried lignin were collected in 2 mL microtubes and dyed with 0.1% CV for 10 min. Then, samples were washed with distilled water until no purple color remained in water. The CV biofilm was solubilized with 30% acetic acid and measured for absorbance at 550 nm using a spectrophotometer (Thermo Scientific, Waltham, MA, USA). Two aliquots of pure organosolv lignin were subject to the same dyeing process and used as blanks.

## 5. Conclusions

We have shown here that lignin is a practical bacterial carrier agent for bioinoculant formulations. The use of lignin as a carrier adds an extra value to bioinoculants since it allows the reincorporation of vegetable organic matter into the soil apart from the microorganisms of interest. In the present study, *A. brasilense* was effectively attached to the lignin surface by biofilm formation, leading to a more reliable immobilization. The next step should be a more extended study on eroded soils colonization and restoration using lignin-immobilized bacteria.

## Figures and Tables

**Figure 1 molecules-24-04613-f001:**
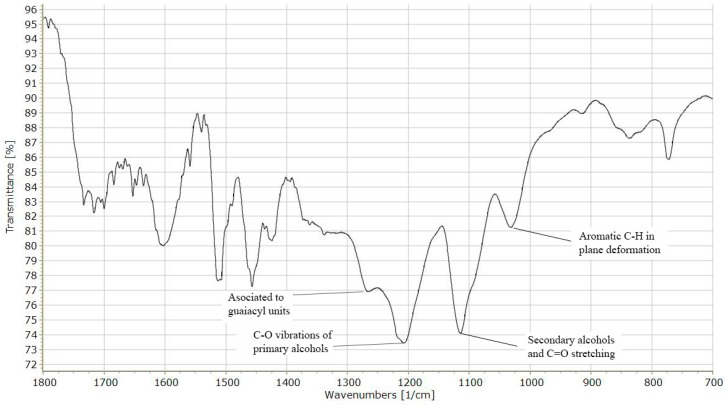
FTIR spectra of the organosolv lignin.

**Figure 2 molecules-24-04613-f002:**
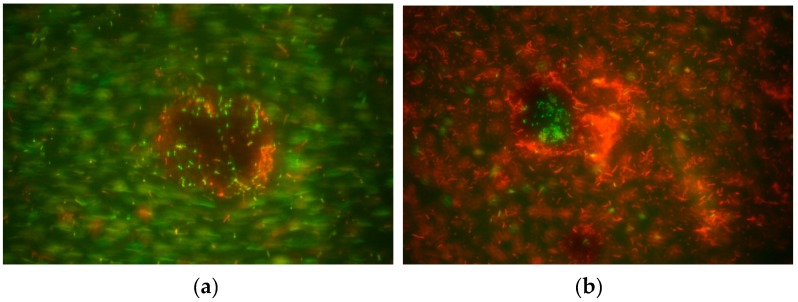
Fluorescent microscope photographs of lignin particles with immobilized *Azospirillum brasilense* cells. (**a**) Free-living moving bacteria in the background and immobilized bacteria adhered to the lignin particle and (**b**) non-living bacteria in the background (orange cells) and living cell bacteria (green cells) immobilized in the lignin particle.

**Figure 3 molecules-24-04613-f003:**
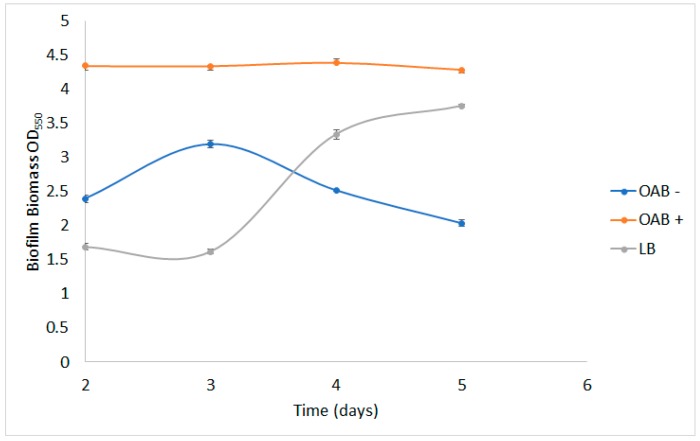
Biofilm formation of *A. brasilense* grown in minimal OAB−, OAB+, and rich medium LB in flasks containing organosolv lignin particles.

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
