# Peer review of "Valorization of Lignin as an Immobilizing Agent for Bioinoculant Production using Azospirillum brasilense as a Model Bacteria"

_molecules, 2019, doi:10.3390/molecules24244613_

Round 1
Reviewer 1 Report
Lignin valorization is an interesting topic. Its use as immobilizing agent could benefit agriculture and other areas.
On other hand, the manuscript contains several deficiencies, which have to be fixed before publication:
English must be revised; Lignin has to be properly characterized. FTIR spectra cannot be direct related with guaiacyl, cumaryl or syringil units. Peaks/bands should be assigned to chemical goups, which can be characteristics of these units. In short, peaks/bands must be properly assigned in the Figure 1 und cited. Figure number are not legible. Figure 2 is meaningless (SEM). Lignin particles have few micrometers of dimension. Structure presented in Figure 2 is not lignin, may be wood, but indeed it is not lignin. Authors have to improve knownledge about lignin and present a proper characterization. FTIR-ATR or transmission? What resolution? Number of scans? What do author mean with "erratic structure of the lignin particles favor the adhesion of the bacterial cells - line 79" . Adhesion is a complex topic and cannot be treated as trivial. Hypothesys has to be proved.Author Response
Dear Reviewer
We regret the number of mistakes in our first submission. We have asked for professional English edition after attend your observations. Hopefuly this time you will find an improved paper and your comments answered.
1) We have described the lignin characterization process as well as the concerning results.
2) We have removed figure 2 as recommended.
3) We removed the hypothesis we formerly planted.
4) The FTIR spectra is explained in more detail.
We appreciate your time an effort for our paper.
Reviewer 2 Report
Nice work. I appreciated the efforts of the authors. I encourage them to keep working on this kind of projects.
In my opinion, the paper should be accepted after some revisions.
Page 2, lines 71-76. The authors should verify their claims about the IR bands attribution and should refer to previous works, since I consider some of the wrong or not very precise. Hence, I suggest to refer to the following papers:
Faix, O (1991) Fourier Transform Infrared Spectroscopy. In Methods of lignin chemistry, edited by Lin, S Y and Dence C W, pp. 83-106. I suggest anybody working with lignin should get this book.
Jahan et al., Bioresource Technology 98 (2007) 465–469.
Savy et al., Biomass & Bioenergy (2014) 58-67.
Page 2, line 73. Please replace “syryngil” with “syringyl”.
Page 2, line 76. Please replace “The lignin particles moropholy is shown inFigure 2” with “The lignin particles morphology is shown in Figure 2”.
I have not commented on the biofilm formation and evaluation, since I have no competences for doing that.
Author Response
Dear Reviewer
We have reviewed and corrected the observed mistakes and asked for English professional edition. Also, your suggestions on the references have been included in this new version. Figure 1 was improved with these suggestions.
We hope this time our work is properly presented. Please receive our gratitude for your time and efforts in reviewing our work.
Regards,
Round 2
Reviewer 1 Report
Accept in present form.